# Combining Self-Reported Information with Radiographic Bone Loss to Screen Periodontitis: A Performance Study

**DOI:** 10.3390/jcm14134531

**Published:** 2025-06-26

**Authors:** José João Mendes, Margarida Neves, Clara Supiot, Leonor Pinto, Diogo Tenda, Nuno Silva, Luís Proença, Yago Leira, Vanessa Machado, João Botelho

**Affiliations:** 1Egas Moniz Center for Interdisciplinary Research (CiiEM), Egas Moniz School of Health & Science, 2829-511 Almada, Portugal; jmendes@egasmoniz.edu.pt (J.J.M.); nunomosilva@hotmail.com (N.S.); lproenca@egasmoniz.edu.pt (L.P.); vmachado@egasmoniz.edu.pt (V.M.); 2Periodontology Unit, Faculty of Medicine and Odontology, University of Santiago de Compostela, 15705 Santiago de Compostela, Spain; yago.leira.feijoo@usc.es

**Keywords:** periodontitis, periodontal disease, prediction model, oral health, radiographic periodontal bone loss, self-report

## Abstract

**Background/Objectives**: The objective of this study is to evaluate the diagnostic performance of a combined screening approach using self-reported periodontal information and radiographic periodontal bone loss (R-PBL) in identifying individuals with periodontitis. **Methods**: An exploratory cross-sectional study was conducted including adult participants with available panoramic radiographs and responses to a validated self-reported periodontal screening questionnaire. R-PBL was assessed on interproximal sites and classified according to established thresholds. Self-reported information followed a validated strategy based on the Center for Diseases Control tool. The performance of individual and combined indicators was analyzed against the 2018 case definition for periodontitis, calculating sensitivity, specificity, positive predictive value (PPV), negative predictive value (NPV), and area under the receiver operating characteristic curve (AUC). **Results**: A total of 150 participants were included, equally divided between periodontitis cases and controls, with a mean age of 46.5 years. The R-PBL model demonstrated the best predictive performance for both periodontitis (AUC: 0.833) and severe periodontitis (AUC: 0.796), with the highest precision and net benefit across thresholds. The Either model showed similar performance, particularly in sensitivity, while SR and Both models underperformed. Decision curve analysis confirmed the superior clinical utility of ‘R-PBL’ and ‘Either’ models in guiding decision-making. **Conclusions**: Combining self-reported information with radiographic bone loss showed adequate screening performance for periodontitis. This dual approach may provide a feasible strategy for identifying high-risk individuals in settings where full clinical examination is not possible.

## 1. Introduction

Periodontitis represents one of the most prevalent chronic inflammatory conditions affecting adults globally [1,2,3]. It is estimated to affect nearly 62% of dentate people, with severe forms affecting nearly 25% of all cases [2]. This disease leads to tooth loss [4], has a detrimental economic effect [5,6], and has been increasingly linked to systemic conditions such as cardiovascular disease, diabetes, and adverse pregnancy outcomes [7]. Also, it affects more socio-economically deprived populations [3,8]. Thus, early screening and detection are crucial for timely intervention and enhanced patient outcomes [5,9,10].

Traditional diagnostic methods for periodontitis rely on clinical examination supplemented with radiographic assessment. Although effective, this diagnosis is time-consuming, operator-dependent, and not always accessible in community-based or resource-limited settings [11]. As a result, predictive models using simplified or digital tools have garnered attention for their potential to identify individuals at risk with greater efficiency and scalability [12,13]. In addition to clinical diagnosis, radiographic periodontal bone loss (R-PBL) (through periapical and/or panoramic X-rays) has been studied in the screening of periodontitis either by clinicians [14] or by machine learning [13,15].

In addition to radiographic screening, self-report tools have also emerged as valuable adjuncts in this triage, particularly in large-scale epidemiological studies and public health settings where clinical assessments may not be feasible [16]. These tools typically consist of structured questionnaires that capture an individual’s perception of their oral health status, symptoms (such as bleeding gums, tooth mobility, or halitosis), and behaviors related to oral hygiene and dental visits [17]. The appeal of self-report tools lies in their scalability, low cost, ease of administration, and capacity for remote or digital deployment [18], but at adequate levels of performance. This is then suited for population-level screening, triage in primary care, or integration into digital health platforms.

The integration of R-PBL with self-report tools may significantly enhance the effectiveness of periodontitis screening, particularly in large-scale or resource-limited settings. By leveraging the strengths of both, there is potential for improved risk stratification and targeted intervention. This study aimed to evaluate and compare the predictive performance and clinical utility of models combining R-PBL and self-reports in identifying individuals with periodontitis and severe periodontitis.

## 2. Materials and Methods

This performance study included a cross-sectional design and is reported following the TRIPOD (Transparent Reporting of a multivariable prediction model for Individual Prognosis Or Diagnosis) statement [19], the checklist for which is available in the Appendix A to guarantee adherence to reporting standards. The Institutional Review Board granted approval for this study (Ethics Committee of Egas Moniz, IDs: 2024/1343 and 2025/1549), which adhered to the principles of the Declaration of Helsinki of 1975, as revised in 2024.

### 2.1. Source of Data and Participants

Data was collected at the Egas Moniz Dental Clinic (EMDC) (Portugal). The participants were incoming first-time patients attending a triage appointment; the study purpose was explained to them, and, then, they were invited to participate. Patient selection followed a consecutive sampling technique at the triage department. After obtaining informed consent, participants completed a questionnaire, and then underwent, in this order, panoramic X-ray (described in Section 2.3|Predictors) and a full-mouth periodontal examination (described in Section 2.2|Outcome).

Patients were included if they were 18 years old or older, able to respond to the questionnaires in Portuguese, and gave consent to participate in the study. Edentulous patients, pregnant or lactating women, or patients with the need of antibiotic prophylaxis before periodontal probing were excluded.

### 2.2. Outcome

Periodontal examination was performed by two trained and calibrated examiners (MN and CS). Intra-class correlation coefficients (ICC) were 0.86 and 0.79 for clinical attachment loss (CAL) and probing depth (PD), respectively. Intra-examiner ICC ranged from 0.90 to 0.85 for both PD and CAL, respectively.

Periodontal diagnosis was based on a full-mouth periodontal examination that included clinical parameters circumferentially using a CP-12 manual periodontal probe (Hu-Friedy^®^, Chicago, IL, USA). At six sites per tooth (mesiobuccal, buccal, distobuccal, mesiolingual, lingual, and distolingual), periodontal pocket depth (PPD), clinical attachment loss (CAL), and bleeding on probing (BOP) were recorded. The definition of these periodontal measures are as follows: PPD—distance from the free gingival margin to the bottom of the pocket; recession (REC)—distance from the cementoenamel junction (CEJ) to the free gingival margin; and a negative sign was assigned if the gingival margin was located coronally to the CEJ. Then, the CAL was calculated as the algebraic sum of the PPD and REC for each site. Measurements were rounded to the nearest whole millimeter. Tooth mobility was assessed according to Lindhe et al. [20], and furcation involvement using a Naber probe [21] and categorized accordingly.

After completing periodontal examination, data was uploaded to a database to categorize the periodontal status. Periodontitis cases were defined according to the American Academy of Periodontology/European Federation of Periodontology 2018 consensus, where a patient was a periodontitis case if interdental CAL was detected in ≥2 mm of non-adjacent teeth, or buccal or oral CAL ≥ 3 mm with pocketing > 3 mm was detected in ≥2 teeth [22]. Each case was then staged and graded accordingly. Severe periodontitis was then categorized as stage III or IV.

### 2.3. Predictors

A self-reported approach was implemented for periodontal status through a validated approach using thirteen questions, of which two had an area under the curve (AUC) of 0.8 of predictive ability [18,23].

Panoramic radiographs were obtained by a certified radiologist using the Viso G5 (Planmeca, Finland) according to the manufacturer’s instructions and stored at an internal cloud through an informatic software system database Romexis (Planmeca, Finland). For X-ray assessment calibration purposes, the same examiners (MN and CS) assessed 10 randomly selected panoramic radiographs and were in complete agreement in 10 of 10 observations (100%).

We then defined four possible screening routes:Confirmation from self-reported and R-PBL data (Both group);Confirmation through self-report (SR group);Confirmation with R-PBL (R-PBL group);Either self-reported or R-PBL confirmation (Either group).

### 2.4. Sample Size

To determine the necessary sample size for assessing our model’s diagnostic accuracy, we conducted a power analysis using the area under the receiver operating characteristic curve (AUC) as a basis. We assumed a null hypothesis AUC of 0.5, indicating no discriminative power, and aimed for a target AUC of 0.8, which suggests good performance. Our calculations identified the minimum number of participants required to achieve statistical significance with a two-sided alpha of 0.05 and a power of 90%. By applying the normal approximation method and converting the AUC to Cohen’s effect size, we found that at least 60 cases of periodontitis would be needed to detect the anticipated effect with the desired statistical power. This balanced design was intended to ensure a reliable evaluation of the model’s ability to discriminate.

### 2.5. Statistical Analysis

Data analyses were conducted using Microsoft Office Excel (Microsoft Corporation, USA), while modeling was performed in R (version 3.1). Four models were evaluated in relation to the organized groups and compared against clinical evaluations. Each model comprised each screening tour:Model Either (or model 1)—Code 1 if positive result for self-reported and R-PBL data; Code 0 if at least one or both of them negative.Model SR (or model 2)—Code 1 if positive result for self-reported; Code 0 if negative result for self-reported.Model R-PBL (or model 3)—Code 1 if positive result for R-PBL; Code 0 if negative result for R-PBL.Model Both (or model 4)—Code 1 if positive result for self-reported OR R-PBL data; Code 0 if both of them showed negative results.

Periodontitis cases were coded as 1, whereas healthy cases were coded as 0. Binary variables were coded as 0 = absence and 1 = presence for all relevant analyses. A subsequent analysis focused on severe periodontitis, with stage III and IV cases coded as 1, and all other cases coded as 0. Sensitivity, specificity, positive predictive value (PPV), and negative predictive value (NPV) were calculated. The area under the curve (AUC) was then computed, accompanied by a 95% confidence interval (CI).

Decision curve analysis (DCA) was utilized to assess and compare the clinical utility of all four models [24]. DCA explicitly includes the clinical consequences of decision thresholds by quantifying the net benefit across a range of threshold probabilities [25,26].

Also, the expected/observed (E/O) ratio was computed. A significance level of *p* < 0.05 was used for all statistical tests.

Formal comparisons between the AUCs of different models were performed using DeLong’s test, as implemented in the ‘pROC’ package, and *p*-values for multiple comparisons were assessed using Bonferroni’s test.

## 3. Results

### 3.1. Participants

The inclusion of participants exceeded the sample calculation due to obtaining healthy patients before reaching the minimum number of periodontitis cases. A total of 75 cases and controls each were obtained. Overall, participants had an average age of 46.5 years old (±19.4), ranging between 18 and 83 years (Table 1). The study population had slightly more women (54.0% of females), and mostly consisted of never smokers (54.7%, n = 82) but with nearly a quarter of active smokers (27.3%, n = 41). Most participants reported having a medium education level (77.3%, n = 116). Following the planned 1:1 ratio, patients living with periodontitis were mostly affected with stage III (22.0%, n = 33) and stage IV (19.3%, n = 29). According to the distribution of grading and extent per staging (Table 2), this sample mostly comprised stage III–IV with a generalized extent and grade B and C.

### 3.2. Models Performance for Periodontitis

In the performance of the models for predicting periodontitis (Table 3), the model with ‘R-PBL’ alone showed the best discriminative ability for predicting periodontitis, with an AUC of 0.833 (95% CI: 0.771–0.892) (Figure 1). It achieved the highest precision (86.7%) among all models. The ‘Either model’ performed similarly, with an AUC of 0.827 (95% CI: 0.767–0.882), sensitivity of 86.7%, and specificity of 78.7%, with an E/O ratio of 1.08 indicating good alignment between predicted and observed cases. The SR and Both models demonstrated weaker performance, with AUCs of 0.613 (95% CI: 0.545–0.680) and 0.620 (95% CI: 0.565–0.675), respectively. DCA plots revealed the ‘R-PBL model’ had the highest net benefit across threshold probabilities, and the ‘Either model’ showed favorable net benefit at moderate thresholds, while SR and Both models offered limited benefits (Figure 2).

### 3.3. Models Performance for Severe Periodontitis

In the performance of the models for predicting severe periodontitis (Table 4), the ‘R-PBL model’ also showed the best discriminative ability for predicting periodontitis, with an AUC of 0.796 (95% CI: 0.724–0.853) (Figure 3). The accuracy, Youden’s index and precision values were lower than for periodontitis prediction models. The ‘Either model’ performed similarly, with an AUC of 0.742 (95% CI: 0.672–0.802), sensitivity of 86.7%, and specificity of 78.7%, with an E/O ratio of 1.08 indicating good alignment between predicted and observed cases. The SR and Both models demonstrated inadequate performance, with AUCs of 0.505 (95% CI: 0.420–0.599) and 0.559 (95% CI: 0.473–0.645), respectively. The clinical usefulness of the models revealed the R-PBL model to provide the highest net benefit across most threshold probabilities, highlighting its utility in guiding decisions around severe periodontitis detection. The Either model also showed favorable net benefit, while SR and Both offered limited clinical value (Figure 4).

DeLong’s test confirmed the results regarding the comparison of models’ AUC curves for periodontitis and severe periodontitis (Table 5).

## 4. Discussion

This study aimed to test the diagnostic performance and clinical utility of four predictive models towards periodontitis and severe periodontitis, using R-PBL and self-reported data alone or combined. In the models evaluated, R-PBL consistently showed superior discriminative performance in both scenarios, achieving AUC values deemed adequate (greater than 0.8). Its balanced sensitivity and specificity, along with the highest net benefit identified in DCA, emphasize its robustness and clinical relevance as a screening tool.

When self-reported measures were included in the predictive models, they did not demonstrate superior performance in their ability to capture subjective signs of disease. The “Either model”, which utilized self-reports with R-PBL, demonstrated high sensitivity and acceptable precision for periodontitis, performing on par with R-PBL across several metrics. These results support previous research indicating that well-designed self-report items can significantly enhance periodontal disease monitoring, especially in populations with limited access to clinical evaluations [17,18]. Although models based solely on simplified rules (SR) or conservative logic (both positive results to self-report and R-PBL) are user-friendly, they exhibited lower diagnostic accuracy and clinical benefit. These models demonstrated reduced sensitivity and Youden’s indices, indicating limited practical utility. Their net benefit was inferior to R-PBL and Either across most threshold probabilities, suggesting that their application may lead to missed cases or unnecessary interventions, contingent upon the clinical context.

The combination of R-PBL with self-reported data seems to present a promising path for scalable screening strategies at the population level, yet using solely R-PBL offers equal or superior performance and clinical utility. By integrating objective clinical guidelines with subjective symptoms and patient-reported behaviors, these hybrid models could enhance early detection and risk assessment. This is particularly significant when considering the global burden of periodontitis, with an estimated 54 billion USD/year in lost productivity and a major portion of the 442 billion USD/year cost for oral diseases [8].

This method aligns with the principles of precision in public health, utilizing accessible data sources to optimize prevention and resource distribution [27]. From a public health standpoint, employing models like R-PBL along with self-reporting could have a significant impact in community settings, primary care, or digital platforms [28]. Their straightforwardness, affordability, and strong performance make them excellent candidates for inclusion in national surveillance systems or targeted screening efforts in populations at risk. Additionally, incorporating DCA into the evaluation framework ensures that the models are assessed not only for statistical accuracy but also for their practical decision-making utility. Yet, our results do not exempt the requirement for clinical confirmation of the person’s periodontal status, as clinical examination is the gold standard. Nevertheless, such an approach could contribute to a more sustainable and accurate oral health primary care response in health systems, but the actual evidence remains limited to a specific clinical setting and requires cautious interpretation.

### Strengths and Limitations

The main advantage of this study is its evaluation framework, which integrates conventional diagnostic performance metrics with decision curve analysis to determine its applicability in clinical settings. The study’s emphasis on the inclusion of self-reports meets the need for scalable screening solutions. However, it is important to acknowledge certain limitations. The models were tested within a specific demographic and context, and further validation in more diverse populations is required to ensure their applicability across different settings [29,30]. Additionally, as a binary model output, this may limit the precision of risk assessments [31]. Future studies shall be conducted to explore further significant risk indicators such as age, smoking habits, education level, living with diabetes, among others. And, while self-reports enhance the model value, they are susceptible to recall bias and subjective interpretation, which could affect the accuracy of predictions [30,32].

We acknowledge that the absence of internal validation (such as cross-validation or bootstrapping) and external validation represents a limitation of this study, which may affect the generalizability of our findings [29,30,31,33,34]. Future research will aim to address this through alternative validation techniques and testing model performance in independent cohorts.

## 5. Conclusions

R-PBL alone or combined with self-reports consistently demonstrated discrimination and clinical utility for predicting periodontitis, but for severe periodontitis, its sole use presents a better performance. Future studies should explore the integration and their performance in larger diverse populations to support personalized and preventive periodontal care.

## Figures and Tables

**Figure 1 jcm-14-04531-f001:**
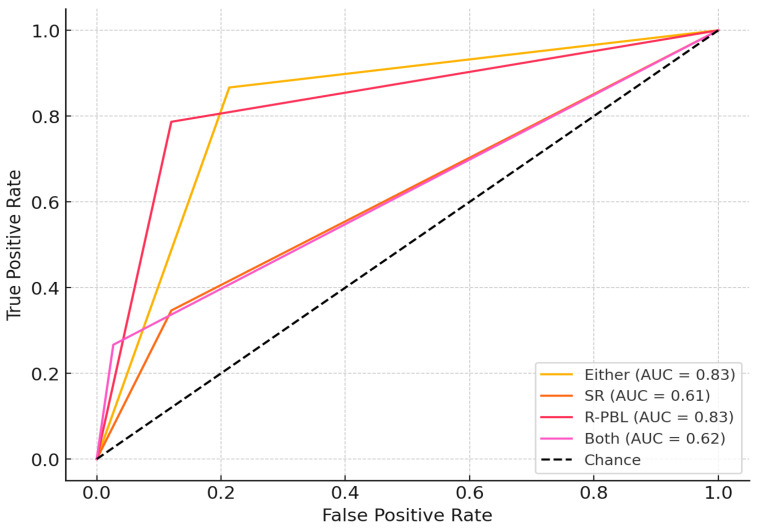
**ROC curve with 95% confidence interval.** ROC curves for the expected outcome of periodontitis for 4 different models (Either, SR, R-PBL and Both) and the dashed gray line is the reference line.

**Figure 2 jcm-14-04531-f002:**
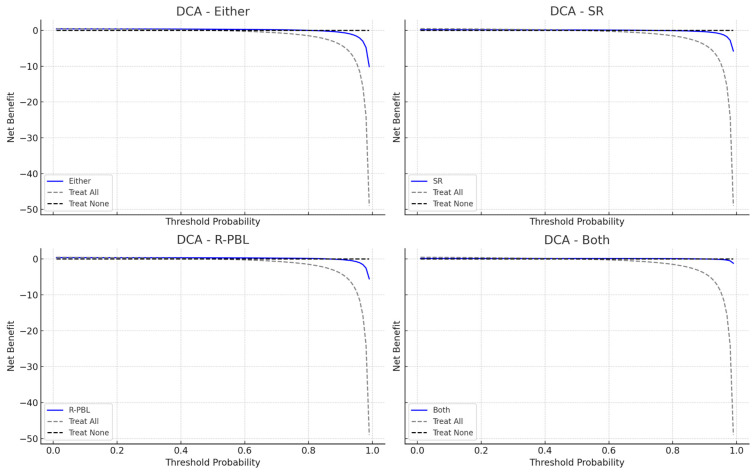
**Decision curve analysis (DCA) plot for each model predicting periodontitis.** The blue line, labeled as the “Expected Outcome Model”, illustrates the net benefit of employing DDS across various probability thresholds. In contrast, the dashed line, marked “Treat All”, indicates the net benefit if everyone is deemed high-risk, while the dotted line, labeled “Treat None”, shows the net benefit if no one is considered high-risk.

**Figure 3 jcm-14-04531-f003:**
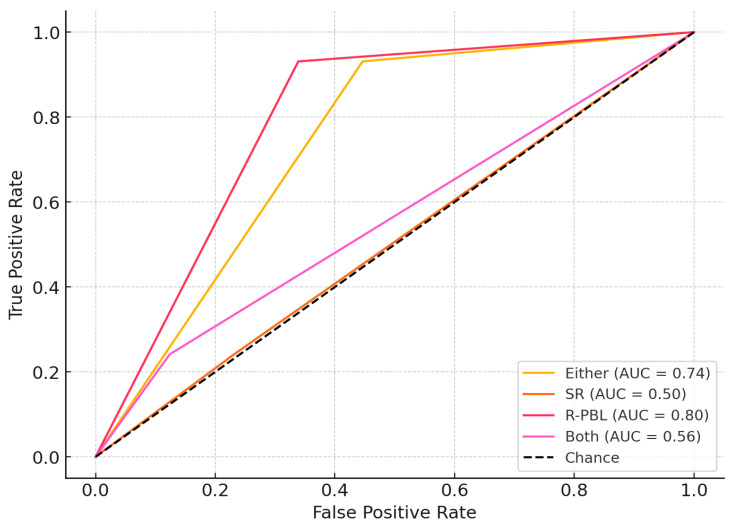
**ROC curve with 95% confidence interval.** ROC curves for the expected outcome of severe periodontitis for 4 different models (Either, SR, R-PBL and Both) and the dashed gray line is the reference line.

**Figure 4 jcm-14-04531-f004:**
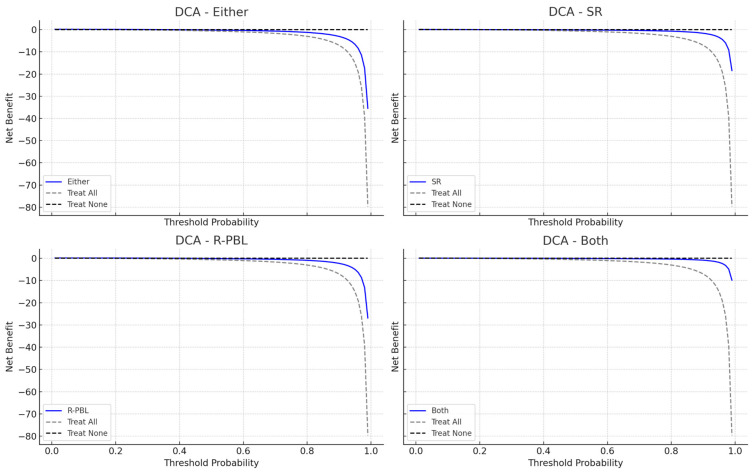
**Decision curve analysis (DCA) plots for each model predicting severe periodontitis.** The blue line, labeled as the “Expected Outcome Model”, illustrates the net benefit of employing DDS across various probability thresholds. In contrast, the dashed line, marked “Treat All”, indicates the net benefit if everyone is deemed high-risk, while the dotted line, labeled “Treat None”, shows the net benefit if no one is considered high-risk.

**Table 1 jcm-14-04531-t001:** Participant characteristics (n = 150). Data is presented as mean and standard deviation (SD) for continuous variables or % and number of cases (n).

Variable	Total
Age, mean (SD) [min–max] (years)	46.5 (19.4) [18–83]
Sex, % (n)	
Female	54.0 (81)
Male	46.0 (69)
Periodontal diagnosis	
Healthy	50.0 (75)
Stage I	4.0 (6)
Stage II	4.7 (7)
Stage III	22.0 (33)
Stage IV	19.3 (29)
Smoking habits	
Never	54.7 (82)
Former	18.0 (27)
Active	27.3 (41)
Education	
Elementary	9.3 (14)
Middle	77.3 (116)
Higher	13.3 (20)

**Table 2 jcm-14-04531-t002:** Distribution of grading and extent of periodontitis according to staging (n = 75).

Staging	Grading			Extent	
	A	B	C	Localized	Generalized
I	1.3 (1)	2.7 (2)	4.0 (3)	1.3 (1)	2.7 (2)
II	1.3 (1)	8.0 (6)	0.0 (0)	2.7 (2)	6.7 (5)
III	5.3 (4)	25.3 (19)	13.3 (10)	6.7 (5)	34.7 (26)
IV	0.0 (0)	16.0 (12)	22.7 (17)	0.0 (0)	37.3 (28)

**Table 3 jcm-14-04531-t003:** Performance analysis for each model in predicting periodontitis.

Model	Sensitivity	Specificity	AUC (95% CI)	E/O Ratio	Accuracy (%)	Youden’s Index (%)	Precision (%)
Either	0.867	0.787	0.827 (0.767–0.882)	1.08	82.7	65.3	80.2
SR	0.347	0.880	0.613 (0.545–0.680)	0.467	61.3	22.7	74.3
R-PBL	0.787	0.880	0.833 (0.771–0.892)	0.907	83.3	66.7	86.7
Both	0.267	0.973	0.620 (0.565–0.675)	0.293	62.0	24.0	90.9

**Table 4 jcm-14-04531-t004:** Performance analysis for each model in predicting severe periodontitis.

Model	Sensitivity	Specificity	AUC (95% CI)	E/O Ratio	Accuracy (%)	Youden’s Index (%)	Precision (%)
Either	0.931	0.553	0.742 (0.672–0.802)	2.79	62.7	48.4	33.3
SR	0.241	0.769	0.505 (0.420–0.599)	1.21	66.7	1.0	20.0
R-PBL	0.931	0.661	0.796 (0.724–0.853)	2.34	71.3	59.2	39.7
Both	0.241	0.876	0.559 (0.473–0.645)	0.76	75.3	11.7	31.8

**Table 5 jcm-14-04531-t005:** Comparison of AUCs using DeLong’s test and adjusted with Bonferroni’s test for multiple comparisons for screening of periodontitis and severe periodontitis.

Model	Either	SR	R-PBL	Both	Either	SR	R-PBL	Both
Either	-	<0.001	0.773	<0.001	-	<0.001	<0.001	<0.001
SR	-	-	<0.001	0.773	-	-	<0.001	<0.001
R-PBL	-	-	-	<0.001	-	-	-	<0.001
Both	-	-	-	-	-	-	-	-

## Data Availability

Data is available upon reasonable request from the corresponding author.

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
