# Peer review of "Combining Self-Reported Information with Radiographic Bone Loss to Screen Periodontitis: A Performance Study"

_jcm, 2025, doi:10.3390/jcm14134531_

Round 1

Reviewer 1 Report

Comments and Suggestions for Authors

Dear authors, your work is very interesting but needs improvement 
The introduction lacks an epidemiological description of periodontitis, how many people are affected worldwide, the epidemiology, whether there are age groups that are more predisposed and the risk factors.
There are no figures to illustrate and explain what your study has focused on, and there are no figures to refer to.
The discussion lacks comparative articles that have already dealt with the same topic in the literature.
The conclusions are weak.
Please improve these parts 

Author Response

We are pleased with the opportunity to revise and resubmit our manuscript titled “Combining self-reported information with radiographic bone loss to screen periodontitis: a performance study” (Manuscript jcm-3680516).

We have considered all comments. Please find appended a track-changes draft of the manuscript and a point-by-point rebuttal to all comments raised as detailed below. Our responses begin with “OUR ANSWER:” in blue-colored text. 

We hope our responses are satisfactory in addressing the criticisms and suggestions.

REVIEWER 1

Dear authors, your work is very interesting but needs improvement 
OUR ANSWER: We thank you for the encouragement. Below you will find our point-by-point response to your remarks. All have been considered thoroughly.

The introduction lacks an epidemiological description of periodontitis, how many people are affected worldwide, the epidemiology, whether there are age groups that are more predisposed and the risk factors.
OUR ANSWER: We added epidemiological description on how many people suffer from periodontitis and expanded a little further on the particularities of periodontitis. Two sentences were added for this purpose:
"It is estimated to affect nearly 62% of dentate people, with severe forms affecting nearly 25% of all cases [2]."
and
"Also, it affects more socio-economically deprived populations [3,7]. "

There are no figures to illustrate and explain what your study has focused on, and there are no figures to refer to.
OUR ANSWER: We appreciate this remark. We were confused on this commentary. We have figures cited and mentioned in the manuscript, particularly pages 5 and 6. We hope this may clarify this reviewer.

The discussion lacks comparative articles that have already dealt with the same topic in the literature.
OUR ANSWER: We appreciate the reviewer’s comment. However, to the best of our knowledge, this is the first study to address this specific topic in this manner. As such, there are no directly comparable articles in the existing literature. We have highlighted this novelty more clearly in the revised discussion.

The conclusions are weak.
OUR ANSWER: We would like to ask this reviewer for more details on how weak and what specific points should be improved. This conclusion was made in accordance with the TRIPOD checklist.

Reviewer 2 Report

Comments and Suggestions for Authors

This manuscript reports a realistic and cost-effective method for screening for periodontal disease, and leads clinical and public health values. The methods are also appropriate. However, revisions as below are required.

#1: lines 136―: Statistical Analysis section

The section seems complicated. Please revise and keep the paragraph simple to improve the reproducibility of the statistical analysis. For example,

 - Please write the binary code (0 or 1) definitions together, not separately.

 - Please provide product information for R and Microsoft Excel.

 - Details of four models should be explained here.

 - A significance level of p-value should be described here.

#2: line 190―: Predictors section

I think the self-reported data inherently have recall and social desirability biases. Describe what you have done to solve the problem, or what you need to do to solve it in the future.

#3: lines 213―: Discussion section

The following article might be cited.

J Clin Periodontol. 2017 May;44(5):456-462.

doi: 10.1111/jcpe.12732.

The article has global prospect, and will enhance this single-center study.

Author Response

We are pleased with the opportunity to revise and resubmit our manuscript titled “Combining self-reported information with radiographic bone loss to screen periodontitis: a performance study” (Manuscript jcm-3680516).

We have considered all comments. Please find appended a track-changes draft of the manuscript and a point-by-point rebuttal to all comments raised as detailed below. Our responses begin with “OUR ANSWER:” in blue-colored text. 

We hope our responses are satisfactory in addressing the criticisms and suggestions.

REVIEWER 2

#1: lines 136―: Statistical Analysis section

The section seems complicated. Please revise and keep the paragraph simple to improve the reproducibility of the statistical analysis. For example,

 - Please write the binary code (0 or 1) definitions together, not separately.

 - Please provide product information for R and Microsoft Excel.

 - Details of four models should be explained here.

 - A significance level of p-value should be described here.

OUR ANSWER: We have written “Binary variables were coded as 0 = absence and 1 = presence for all relevant analyses.”. We hope this satisfies this reviewer. We provided product information for R and Microsoft Excel. We have detailed the models and each binary coding. At last, we added a sentence clarifying significance level of p-value. The updated section now reads as follows:

Data analyses were conducted using Microsoft Office Excel (Microsoft Corporation, USA), while modeling was performed in R (version 3.1). Four models were evaluated in relation to the organized groups and compared against clinical evaluations. Each model comprised each screening tour:

  • Model Either (or model 1) – Code 1 if positive result for self-reported and R-PBL data; Code 0 if at least one or both of them negative;
  • Model SR (or model 2) – Code 1 if positive result for self-reported; Code 0 if negative result for self-reported;
  • Model R-PBL (or model 3) – Code 1 if positive result for R-PBL; Code 0 if negative result for R-PBL;
  • Model Both (or model 4) – Code 1 if positive result for self-reported OR R-PBL data; Code 0 if both of them showed negative results.

Periodontitis cases were coded as 1, whereas healthy cases were coded as 0. Binary variables were coded as 0 = absence and 1 = presence for all relevant analyses. (…)”

#2: line 190―: Predictors section

I think the self-reported data inherently have recall and social desirability biases. Describe what you have done to solve the problem, or what you need to do to solve it in the future.

OUR ANSWER: We fully acknowledge that self-reported data are subject to inherent limitations such as recall bias and social desirability bias. While these issues could not be fully addressed in the present study, we plan to mitigate them in future research by complementing self-reported data with objective measures whenever possible, employing validated questionnaires with proven reliability, and incorporating methodological strategies (e.g., anonymous data collection, repeated measures) to reduce bias. By acknowledging this, the original submission already included a sentence addressing this: “And, while self-reports enhance the model value, they are susceptible to recall bias and subjective interpretation, which could affect the accuracy of predictions [27,28].”

#3: lines 213―: Discussion section

The following article might be cited.

J Clin Periodontol. 2017 May;44(5):456-462.

doi: 10.1111/jcpe.12732.

The article has global prospect, and will enhance this single-center study.

OUR ANSWER: We have added a sentence that cites this article accordingly. Now reads as follows: “This is particularly significant when considering the global burden of periodontitis, with an estimated 54 billion USD/year in lost productivity and a major portion of the 442 billion USD/year cost for oral diseases [22].”

Reviewer 3 Report

Comments and Suggestions for Authors
  • In the methods section, periodontitis cases were coded as 1, while healthy 139 cases were coded as 0. But Tables 1 and 2 describe 4 stages of periodontitis and the classification and extent. 
  • Tables 3 and 4 show the results for four groups: Either, SR, R-PBL, and Both. Clarify what it means? 

Author Response

We are pleased with the opportunity to revise and resubmit our manuscript titled “Combining self-reported information with radiographic bone loss to screen periodontitis: a performance study” (Manuscript jcm-3680516).

We have considered all comments. Please find appended a track-changes draft of the manuscript and a point-by-point rebuttal to all comments raised as detailed below. Our responses begin with “OUR ANSWER:” in blue-colored text. 

We hope our responses are satisfactory in addressing the criticisms and suggestions.

REVIEWER 3

In the methods section, periodontitis cases were coded as 1, while healthy cases were coded as 0. But Tables 1 and 2 describe 4 stages of periodontitis and the classification and extent. 

OUR ANSWER: Tables 1 and 2 were included to provide a detailed description of participant characteristics, in line with the TRIPOD checklist. While the models used binary coding for periodontitis status (1 = periodontitis, 0 = healthy), we presented the distribution of periodontitis stages and extent according to the 2018 case definition to offer greater clinical context and transparency regarding the study population.

Tables 3 and 4 show the results for four groups: Either, SR, R-PBL, and Both. Clarify what it means? 

OUR ANSWER: We have revised and improved the statistical analysis section to clarify the meaning of the groups presented in Tables 3 and 4. We hope this is now clearer and remain happy to further revise if the reviewer finds that additional clarification is needed.

Reviewer 4 Report

Comments and Suggestions for Authors

Dear Authors, thank you for the opportunity to review this manuscript. The aim of this study is to evaluate the diagnostic performance of a combined screening approach using self-reported periodontal information and radiographic periodontal bone loss (R-PBL) in identifying individuals with periodontitis. The topic is interesting, but there are some important inaccuracies.

  1. The method of patient selection is not clearly specified: it is unclear whether selection was consecutive, randomized, or otherwise, which may introduce potential selection bias. In addition, explicit inclusion and exclusion criteria are not reported.

  1. The statistical analysis does not include formal comparisons between the AUCs of the different models (e.g., DeLong’s test), which makes the statements regarding the superiority of one model over another less supported. It would be advisable either to perform such a comparison or to explain why it was not conducted.

  1. The methodological section provides a detailed description of the clinical assessment (periodontal measures), but does not sufficiently elaborate on key aspects related to the construction and evaluation of the predictive models. Expanding this part would enhance the methodological transparency of the study

  1. The probability thresholds used in the Decision Curve Analysis (DCA) are neither justified nor discussed, and it is not explained how these thresholds were selected to reflect realistic clinical scenarios. Providing such a rationale would enhance the interpretability and practical relevance of the DCA findings

  1. The developed models were not externally validated, nor was a rigorous internal validation approach (such as cross-validation or bootstrapping) applied, which considerably limits the generalizability of the results. Addressing this point would strengthen the robustness of the study’s conclusions

  1. The emphasis on potential applications in public health and digital platforms appears somewhat overstated in relation to the actual evidence provided, which remains limited to a specific clinical setting. A more cautious interpretation in this regard would be advisable

  1. The 'Strengths and Limitations' section is rather limited and does not critically address several important methodological limitations mentioned above. Expanding this section with a more balanced discussion would improve the transparency and rigor of the manuscript.

  1. The text contains some repetitions and redundancies, particularly in the description of performance metrics, and could be streamlined to improve clarity and readability

  1. It would be helpful to indicate in the header of Table 1 how the data are presented (e.g., [% (n)]), to improve clarity for the reader.

  1. Regarding the self-reported questionnaire, the 13 items used are not specified, nor are their individual results reported (e.g., in a table or as supplementary material). Additionally, the scoring strategy is not clearly defined: it is unclear how the 13 responses were combined to yield a 'positive/negative' classification for the model. Furthermore, for the self-reported model alone, it is not reported which items were the most informative, unless a total score was used, which would seem inconsistent with the statement that two specific items had an AUC > 0.8 (though these items are not identified). Clarifying these aspects would greatly enhance the transparency and reproducibility of the study.

  1. t is also unclear whether the self-reported questionnaire was administered by an interviewer or self-administered. If self-administered, it would be important to specify whether it was provided in the participants’ native language and whether a validated version was used. If this information is not available, it should be acknowledged as a limitation

  1. The term 'predictive model' in the present study is used in a broad sense, as the authors implemented predefined decision rules based on radiographic bone loss (R-PBL) and self-reported periodontal information, rather than developing a true multivariable statistical model (e.g., logistic regression). This approach provides an easily applicable screening strategy but does not explore the potential contribution of other relevant variables (such as smoking, age, or education level), which were collected but not included in the predictive process. It would be appropriate to acknowledge this point in the limitations section, as including additional variables could have improved the model's predictive performance and allowed for a more comprehensive risk stratification.

Author Response

We are pleased with the opportunity to revise and resubmit our manuscript titled “Combining self-reported information with radiographic bone loss to screen periodontitis: a performance study” (Manuscript jcm-3680516).

We have considered all comments. Please find appended a track-changes draft of the manuscript and a point-by-point rebuttal to all comments raised as detailed below. Our responses begin with “OUR ANSWER:” in blue-colored text. 

We hope our responses are satisfactory in addressing the criticisms and suggestions.

REVIEWER 4

Dear Authors, thank you for the opportunity to review this manuscript. The aim of this study is to evaluate the diagnostic performance of a combined screening approach using self-reported periodontal information and radiographic periodontal bone loss (R-PBL) in identifying individuals with periodontitis. The topic is interesting, but there are some important inaccuracies.

  1. The method of patient selection is not clearly specified: it is unclear whether selection was consecutive, randomized, or otherwise, which may introduce potential selection bias. In addition, explicit inclusion and exclusion criteria are not reported.

OUR ANSWER: Thank you for this remark. We have improved the 2.1 section to address this commentary, that now reads as follows: “Data was collected at the Egas Moniz Dental Clinic (EMDC) (Portugal). Participants were first incoming patients attending a triage appointment, and the study purpose was explained to them and then they were invited to participate. Patient selection followed a consecutive sampling technique at the Triage department. Following informed consent signature, participants completed a questionnaire, and then underwent, on this order, panoramic x-ray (described in section 2.3 | Predictors) and a full-mouth periodontal examination (described in section 2.2 | Outcome).

Patients were included if: were 18 years old or older; able to respond to the questionnaires in Portuguese; consent to participate in the study. Edentulous patients, pregnant or lactating women, or patients with the need of antibiotic prophylaxis before periodontal probing.”

  1. The statistical analysis does not include formal comparisons between the AUCs of the different models (e.g., DeLong’s test), which makes the statements regarding the superiority of one model over another less supported. It would be advisable either to perform such a comparison or to explain why it was not conducted.

OUR ANSWER: We thank the reviewer for this valuable suggestion. We have now performed formal comparisons between the AUCs of the different models using DeLong’s test, implemented via the pROC package in R. The results of these comparisons have been included in the revised Results section.

  1. The methodological section provides a detailed description of the clinical assessment (periodontal measures), but does not sufficiently elaborate on key aspects related to the construction and evaluation of the predictive models. Expanding this part would enhance the methodological transparency of the study

OUR ANSWER: We expanded on the construction and evaluation of the predictive models, particularly on the statistical analysis on how we computed more in detail the binary categorization.

The probability thresholds used in the Decision Curve Analysis (DCA) are neither justified nor discussed, and it is not explained how these thresholds were selected to reflect realistic clinical scenarios. Providing such a rationale would enhance the interpretability and practical relevance of the DCA findings

OUR ANSWER: We expanded on why DCA was relevant to use, particularly we added: “DCA explicitly includes the clinical consequences of decision thresholds by quantifying the net benefit across a range of threshold probabilities.”

The developed models were not externally validated, nor was a rigorous internal validation approach (such as cross-validation or bootstrapping) applied, which considerably limits the generalizability of the results. Addressing this point would strengthen the robustness of the study’s conclusions

OUR ANSWER: We fully acknowledge this important limitation. As this was an exploratory study primarily aimed at developing and comparing candidate models, external validation was beyond its scope. In addition, while internal validation via cross-validation or bootstrapping was not initially performed, we recognize that applying such approaches would enhance the robustness and generalizability of the findings. We have now addressed this limitation explicitly in the Discussion and plan to prioritize rigorous internal and external validation in future work.

The emphasis on potential applications in public health and digital platforms appears somewhat overstated in relation to the actual evidence provided, which remains limited to a specific clinical setting. A more cautious interpretation in this regard would be advisable

OUR ANSWER: We understand this remark. We acknowledge this by adding the following sentence before section 4.1.: “(…) but the actual evidence remains limited to a specific clinical setting and requires cautious interpretation.”

The 'Strengths and Limitations' section is rather limited and does not critically address several important methodological limitations mentioned above. Expanding this section with a more balanced discussion would improve the transparency and rigor of the manuscript.

OUR ANSWER: Due to the reviewers’ comments, this section was expanded and improved.

The text contains some repetitions and redundancies, particularly in the description of performance metrics, and could be streamlined to improve clarity and readability

OUR ANSWER: We hope this revised document may have less repetitions and redundancies. We are happy to further address this point, if the reviewer’s still finds it relevant.

It would be helpful to indicate in the header of Table 1 how the data are presented (e.g., [% (n)]), to improve clarity for the reader.

OUR ANSWER: We have completed Table 1 header as follows: “Table 1. Participants characteristics (n=150). Data is presented as mean and standard deviation (SD) for continuous variables or % and number of cases (n).”

Regarding the self-reported questionnaire, the 13 items used are not specified, nor are their individual results reported (e.g., in a table or as supplementary material). Additionally, the scoring strategy is not clearly defined: it is unclear how the 13 responses were combined to yield a 'positive/negative' classification for the model. Furthermore, for the self-reported model alone, it is not reported which items were the most informative, unless a total score was used, which would seem inconsistent with the statement that two specific items had an AUC > 0.8 (though these items are not identified). Clarifying these aspects would greatly enhance the transparency and reproducibility of the study.

OUR ANSWER: We appreciate this point. The questionnaire has 13 items, however the items used for self-reported periodontitis followed the validated protocol in Machado, V.; Botelho, J.; Proença, L.; Mendes, J.J. Self-Reported Illness Perception and Oral Health-Related Quality of Life Predict Adherence to Initial Periodontal Treatment. J Clin Periodontol 2020, 47, 1209–1218, doi:10.1111/jcpe.13337. We emphasize this reference is number 20, already mentioned in the manuscript submitted. We strictly followed this protocol, and to avoid redundancies, we opted to cite the original article with the translated and validated instrument.

It is also unclear whether the self-reported questionnaire was administered by an interviewer or self-administered. If self-administered, it would be important to specify whether it was provided in the participants’ native language and whether a validated version was used. If this information is not available, it should be acknowledged as a limitation

OUR ANSWER: We followed the original protocol questionnaires were provided to the participants, they had to understand and speak Portuguese and questionnaire was validated as cited accordingly in the submitted article.

The term 'predictive model' in the present study is used in a broad sense, as the authors implemented predefined decision rules based on radiographic bone loss (R-PBL) and self-reported periodontal information, rather than developing a true multivariable statistical model (e.g., logistic regression). This approach provides an easily applicable screening strategy but does not explore the potential contribution of other relevant variables (such as smoking, age, or education level), which were collected but not included in the predictive process. It would be appropriate to acknowledge this point in the limitations section, as including additional variables could have improved the model's predictive performance and allowed for a more comprehensive risk stratification.

OUR ANSWER: We thank the reviewer for this insightful comment. We fully agree that the term 'predictive model' was used in a broad sense to reflect the predefined decision rules based on radiographic bone loss and self-reported periodontal information, rather than a multivariable statistical model. We acknowledge that incorporating additional relevant variables, such as smoking, age, and education level, could potentially enhance predictive performance and support more comprehensive risk stratification. We have now explicitly addressed this point in the limitations section and will consider this direction in future model development. The added new sentence to section reads as follows: “Future studies shall be conducted to explore further significant risk indicators such as age, smoking habits, education level, living with diabetes, among others”.

Round 2

Reviewer 1 Report

Comments and Suggestions for Authors

Dear Authors, by following my advice, you have significantly improved the manuscript.
Congratulations.
I am not requesting any further changes.

Author Response

We appreciate the carefulness

Reviewer 2 Report

Comments and Suggestions for Authors

Thank you for submitting the revised version of your manuscript. The revisions improve the clarity and overall quality of the paper.

Author Response

We appreciate the carefulness

Reviewer 3 Report

Comments and Suggestions for Authors

Now it can be accepted

Author Response

We appreciate the carefulness

Reviewer 4 Report

Comments and Suggestions for Authors

Thank you for the opportunity to review this revised version of your manuscript. The aim of the study, to evaluate the diagnostic performance of a combined screening approach using self-reported periodontal information and radiographic periodontal bone loss (R-PBL), addresses an important and timely topic in periodontal diagnostics. You have made substantial improvements to the manuscript, and the revisions have clearly enhanced both the clarity and methodological transparency of the study. I also appreciate that you have provided clear and thoughtful responses to all the comments raised during the initial review. However, there are still two specific aspects that remain insufficiently clarified and would benefit from further elaboration:

Point 4: The referenced study appears to employ two well-established tools, the Brief Illness Perception Questionnaire (Brief-IPQ) and the Oral Health Impact Profile (OHIP-14), which contain 9 and 14 items, respectively. These instruments are primarily designed to assess illness perception and oral health-related quality of life, rather than for direct screening or diagnostic classification of periodontitis. The study does not include or validate a specific 13-item screening tool for periodontitis, as suggested in your response. In light of this, it remains unclear what the 13 items used in your model are; How these items were scored or combined to generate a binary classification (positive/negative); Which two items achieved an AUC > 0.8, and on what basis this was determined. Providing this information, ideally in the main text or as supplementary material, would significantly improve the clarity and scientific value of your work.

Point 10: While the added explanation about the general utility of DCA is appreciated, it does not fully address the core concern of the original comment. Specifically, the justification for the choice of threshold probabilities remains insufficient. The authors mention that net benefit was assessed across a range of thresholds, but it is still unclear why that specific range was selected, and whether it reflects realistic clinical decision-making scenarios in periodontal screening. For instance, thresholds commonly used in public health triage or based on prevalence-adjusted risk tolerance could have been cited or discussed. I encourage the authors to either clarify the rationale for the chosen threshold range or cite relevant literature supporting these thresholds in similar screening studies.

Clarifying these final points would further strengthen the manuscript and ensure its full scientific transparency and replicability.

Author Response

We are pleased with the opportunity to revise and resubmit our revised manuscript titled “Combining self-reported information with radiographic bone loss to screen periodontitis: a performance study” (Manuscript jcm-3680516.R1).

Once again, we have considered all comments. Please find appended a track-changes draft of the manuscript and a point-by-point rebuttal to all comments raised as detailed below. Our responses begin with “OUR ANSWER:” in blue-colored text. 

We hope our responses are satisfactory in addressing the criticisms and suggestions.

REVIEWER 4

Thank you for the opportunity to review this revised version of your manuscript. The aim of the study, to evaluate the diagnostic performance of a combined screening approach using self-reported periodontal information and radiographic periodontal bone loss (R-PBL), addresses an important and timely topic in periodontal diagnostics. You have made substantial improvements to the manuscript, and the revisions have clearly enhanced both the clarity and methodological transparency of the study. I also appreciate that you have provided clear and thoughtful responses to all the comments raised during the initial review. However, there are still two specific aspects that remain insufficiently clarified and would benefit from further elaboration:

OUR ANSWER: Thank you once again for the thorough review process and for acknowledging our efforts to respond to and resolve all remarks you have raised.

Point 4: The referenced study appears to employ two well-established tools, the Brief Illness Perception Questionnaire (Brief-IPQ) and the Oral Health Impact Profile (OHIP-14), which contain 9 and 14 items, respectively. These instruments are primarily designed to assess illness perception and oral health-related quality of life, rather than for direct screening or diagnostic classification of periodontitis. The study does not include or validate a specific 13-item screening tool for periodontitis, as suggested in your response. In light of this, it remains unclear what the 13 items used in your model are; How these items were scored or combined to generate a binary classification (positive/negative); Which two items achieved an AUC > 0.8, and on what basis this was determined. Providing this information, ideally in the main text or as supplementary material, would significantly improve the clarity and scientific value of your work.

OUR ANSWER: We apologize for this typo. The system used, Zotero, linked to a wrong citation. We have corrected this, and now links to the correct reference: Machado, V.; Lyra, P.; Santos, C.; Proença, L.; Mendes, J.J.; Botelho, J. Self-Reported Measures of Periodontitis in a Portuguese Population: A Validation Study. J Pers Med 2022, 12, 1315, doi:10.3390/jpm12081315. To avoid duplication, and possibly issues with authentication, we opted to not include the items, as they are clearly detailed in the article cited. We understand that prior to this typo acknowledgment, it was unclear which items were referring to, but now it is clear and transparent on the original article cited.

Point 10: While the added explanation about the general utility of DCA is appreciated, it does not fully address the core concern of the original comment. Specifically, the justification for the choice of threshold probabilities remains insufficient. The authors mention that net benefit was assessed across a range of thresholds, but it is still unclear why that specific range was selected, and whether it reflects realistic clinical decision-making scenarios in periodontal screening. For instance, thresholds commonly used in public health triage or based on prevalence-adjusted risk tolerance could have been cited or discussed. I encourage the authors to either clarify the rationale for the chosen threshold range or cite relevant literature supporting these thresholds in similar screening studies.

OUR ANSWER: Thank you for your remark. We have cited the two references regarding the thresholds, and hope this adding may clarify this point accordingly to your suggestion “cite relevant literature supporting these thresholds in similar screening studies”. These articles are highly cited and reference articles regarding this threshold range.